# The Definition of Play: A Measurement Scale for Well-Being Based on Human Physiological Mechanisms

Yoshihiro Shimomura 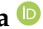

Design Research Institute, Chiba University, Chiba 2638522, Japan; shimomura@faculty.chiba-u.jp

**Abstract:** Play is an activity common to all cultures and is thought to be a useful way to improve well-being since it brings about enjoyment. This study aimed to comprehensively define play and develop a method to evaluate what types of play lead to well-being. It defined play as "the activation of one's reward system through intrinsically motivated decisions and actions of the self, not for the direct purpose of survival", based on human physiology, including brain science relating to motivation and behaviour. It checked this definition by conducting an online survey and applying a measurement scale to quantify the degree of play. The results revealed that the favourite play activity involved a significantly higher degree of play than the highest-effort habits (defined as the activities that participants most disliked but continued to take part in). There was also a significant difference in intrinsic motivation, reward system activation, and decision-making but no difference in action of the self. This method made it possible to evaluate the intensity of each element of the brain mechanism making up play, and it can, therefore, be used to examine the well-being of groups, organisations, and individuals across domains.

**Keywords:** well-being; play; enjoyment; physiological definition; measurement

## 1. Introduction

One activity common to all cultures, regardless of country or domain, is play. Play and the resultant enjoyment that is derived from it may be important for well-being, including daily life, work–life balance, teaching and learning, and developing and maintaining social connections [1]. Therefore, increasing play and the subsequent enjoyment should improve well-being. However, since enjoyment is one of the emotions experienced as a result of various activities, it is difficult to specify how it can be used to promote well-being. As such, our study focused on play as a typical activity that provides enjoyment. Because, as noted above, play is a common practice among human beings [2], we believe that play may be a useful instrument for improving well-being.

Well-being is referred to in the United Nations Sustainable Development Goals (SDGs) in Goal 3, which is titled "Ensure healthy lives and promote well-being for all at all ages" [3], which is the philosophy of the World Health Organization (WHO) [4]. In general, the role of enjoyment in well-being is taken for granted and is not emphasised as much as safety, health, or the economy. Therefore, it is necessary to review previous studies relating to well-being to examine what leads to enjoyment. In recent years, a number of studies have been conducted in various domains, such as loneliness during the pandemic [5], working style [6], housing and urban planning [7], transportation [8], socio-economics [9], early childhood education [10], school [11], higher education [12], young people [13], and the elderly [14]. For example, in a 2015 study by Weziak-Bialowolska, the concept of well-being was interpreted as a set of domains comprising emotional health, physical health, meaning and purpose, social connections, character strength, and financial stability [15]. Regarding the domain of social welfare, López-Concepción [9] stated that trust in institutions (large corporations, governments, banks) as well as science and technology are making human life healthier, easier, and more comfortable. In Pasek's 2022 study, it

was noted that from a psychological perspective, physical activity, along with self-esteem, is assumed to be associated with well-being [11]. Kim (2023) argued that psychological well-being is important for older adults' quality of life [14]. The enjoyment expected to be experienced from such domains will vary from country to country and person to person. While recognising this diversity, we must strive to promote and enhance well-being. In this study, we believe that it is necessary not to focus on differences in domains but to focus on how enjoyment can be used to promote well-being regardless of domain. Therefore, we sought to investigate what aspects of play are common to all human societies.

The purpose of this study was to propose a comprehensive definition of play, as well as to develop a method to evaluate what types of play lead to the promotion of well-being. This evaluation method was intended to be used as a tool to help consider the well-being of people in groups and organisations by quantifying the degree of play and for individuals to conduct self-evaluation. By making it possible to consider well-being through cross-domain concepts, we hope to create a world that as many people as possible can enjoy.

## 2. Materials and Methods

### 2.1. Literature Review

Play is generally defined as an activity that is spontaneous, fun, and has an end in itself. It is also considered advantageous for acquiring social skills and physical skills, but it is not considered to be work or serious. In the field of social sciences, Huizinga attempted to define the elements of play in culture in 1955 [2]. He argued that human culture was established and developed as play in itself. Based on a vast number of historical facts, he examined play in various human behaviours, such as words, music, martial arts, and gambling. Huizinga's discussion of play made it a popular subject of study. Lieberman (1977) stated that play comprised the following five elements: physical spontaneity, manifest joy, sense of humour, social spontaneity, and cognitive spontaneity [16]. Barnett (1990) focused on expanding on Lieberman's definition of play and developed a playfulness scale for children [17]. Therefore, the recent research has focused on the relationship between play and well-being. According to Proyer (2013), playfulness in adults has a strong positive relationship with life satisfaction, a tendency toward pleasurable activities, and an active lifestyle [18]. Play is not only for infants and children but is found to be associated with well-being in various age groups [19]. For example, academic performance, anxiety, and bullying were all found to relate to levels of playfulness in adolescence and, in turn, overall well-being. [20]. Even in psychiatry, play is receiving more attention, and its relationship with well-being was investigated [21]. In the field of sustainability, several studies were conducted on the topic of play. It was resolved that children in urban settings have the right to play in an environment with clean air and that this needs to be protected for a sustainable future [22]. The importance of early childhood play, that is, play that expresses the fictional world, was clarified to be a place where one may learn about social Sustainability [23]. Furthermore, in the field of anthropology, play in primates that are closely related to humans is being studied [24]. In primates other than humans, a subjective evaluation cannot be performed, therefore, research was conducted based on behavioural observations. In this domain, re-ordered, exaggerated, repetitive, fragmented, and incomplete activities are considered to be play.

Thus, there has been a considerable amount of research conducted on play, which is rich in variation and scope. However, no unified context exists for this topic. Each researcher has adopted a different approach to whether they will rely on an initial definition or, more recently, on a definition that includes aspects of well-being, or on their own definition. In addition, these methodologies each consider the researcher's specific area of knowledge first, and then, play is defined in relation to that domain. This may make it difficult to consider well-being regardless of domain. For example, elements such as spontaneous behaviours and humorous conversations that are thought to be present in play are typified by consciousness and the behaviours expressed. Since the types of expression vary depending on the culture in each country, the presence or absence of

physical disabilities, and generations, it is necessary to explain the consistency each time a result is applied to another domain. As a result, the viewpoint of whether the expression type is essential is ignored. In addition, the explanation that play is not work and that it is not serious is dependent on the context and is not essential. In other words, if the features of work or seriousness are present, play can still be possible. With respect to the explanation that it is advantageous in acquiring sociality and skills, these can be said to be the effects obtained as a result of play, not the reason for play itself. Proyer stated, "adult playfulness is an individual difference variable that allows people to (re)frame everyday situations in an interesting, exciting, and entertaining way" [25]. While these are also the results of brain and whole-body functions, Proyer's explanation is limited by the fact that playfulness is one of the individual characteristics. This definition does not capture play itself but rather reflects an aspect of human nature. Traits such as being other-directed and whimsical, along with their associated behaviours, represent just one manifestation of its essence. It seems that the internal processes within humans that give rise to play have not yet been explained. In fact, the study of play has not yet eliminated this ambiguity. This is because the meaning of play itself for humans has not yet been thoroughly studied.

### 2.2. Review of Play Evaluation Methods

A five- or seven-point Likert-type scale is typically used to assess the degree of expression of play-related behaviours and consciousness. Barnett [16] developed a scale consisting of 23 items, with sub-items, to measure the five constituent elements defined by Lieberman [17]. For example, the sub-item of the element "Physical spontaneity" was assessed using "The child is physically active during play"; "The child shows enthusiasm during play" was utilised to measure "Manifest Joy", while "The child enjoys joking with other children" was employed to evaluate "Sense of Humour." The Short Measure of Adult Playfulness (SMAP) consisting of five items, the 32-item Adult Playfulness Scale (APS), and the Orientation to Happiness Scale (OTH), comprising 18 items, are also used to measure playfulness [26–28]. Proyer proposed a structural model of adult playfulness comprising four facets and 28 items, namely, Other-directed, Light-hearted, Intellectual, and Whimsical (OLIW scale) [20,29]. Subsequently, the above scale was refined to a 12-item short form scale (the OLIW-S) [25]. In the area of sustainability, there are no studies that directly make use of scales assessing play, although several well-being scales have been employed, including meanings close to play evaluation scales. For example, Lorber used loneliness, life satisfaction, and mental health as indicators [5]. Weziak-Bialowolska evaluated the relationships between the perceived importance of six well-being domains, including meaning and purpose, social connectedness, emotional health, character strengths, physical health, and financial stability [15]. Yan evaluated kindergarten teachers' happiness based on job satisfaction, life satisfaction, and personal growth [10].

Thus, many methods of evaluation were proposed, and aspects related to play and well-being were measured numerically to a certain extent. However, these studies measured the results of play, such as expressed consciousness and behaviour, through subjective observations and did not measure the play mechanism. To date, no method has been proposed for measuring the source of these expressions of enjoyment.

### 2.3. Definition of Play
#### 2.3.1. Biological Basis

It is necessary to elucidate what play means to human beings, and upon investigation, we gained insight that the brain is the source of the expression of behaviour related to play. Perhaps the most important related mechanisms are those of motivation and behaviour. Motivation as a mechanism to bring about play and enjoyment does not contradict Huizinga's theory of play as a driving force of culture, Barnett's constituent elements of spontaneity, or Proyer's sense of fun, stimulation, and interest [2,17,25]. To clarify the meaning of play, previous studies focusing on the function of the human brain were examined from the viewpoint of motivation and behaviour.

Motivation is defined as the process whereby goal-directed activity is instigated and sustained [30]. Motivation can be intrinsic or extrinsic. When an individual engages in intrinsic motivation, the brain's reward system is activated [31]. The reward system is a dopaminergic nervous system network that extends from the ventral tegmental area of the midbrain to the prefrontal cortex. Therefore, when goal-oriented activities are carried out spontaneously, the reward is pleasure. Extrinsic motivation attenuates the activity of the striatum, which forms part of the decision-making and motor command systems [31]. This is known as the undermining effect. For example, what was originally a fun hobby becomes less enjoyable and an individual is less motivated to engage in such a hobby when an external financial reward is offered for engaging in the hobby. The dorsal and ventral portions of the striatal putamen are associated with long- and short-term reward predictions, respectively [32]. Therefore, the concept of time is important in reward prediction. Cognitive reappraisal can lead to the reassessment of demands, transforming a feared task into a challenge, leading to positive emotions and happiness [33]. This indicates that the reward can be increased by the will. Additionally, this suggests that overcoming difficulties during play increases satisfaction with play. The nucleus accumbens, which forms part of the reward system, is activated by physiological or psychological signals [34]. Based on these characteristics of the brain, it is thought that intrinsic motivation that is associated with several benefits, including enjoyment, is the source of the motivation for play. Regardless of whether what gives the sensation is real or not, even if it is fiction, the establishment of reward suggests the essence of play.

Behaviour and expressed consciousness are the two most significant aspects of play research. Therefore, similar to an examination into motivation, the mechanisms in the brain linked with behaviour must be investigated. It is important to note that there are no reward-system-specific receptors for various behaviours. Dopamine is associated with the activity of specific nuclei, such as the nucleus accumbens, putamen, and ventral tegmental area, to establish a hedonic element in the reward system [35]. In other words, a single rewards system exists and it is, therefore, difficult to determine whether certain behaviour will be perceived as a reward by all individuals. For example, individuals who can move their bodies the way they want to may enjoy sports as play, while those who do not share these abilities may not engage sports as a form of play. This highlights that the exercises and tools attributed to play are only suitable for those who are able to use them. It is well established that the striatum receives input not only from the ventral tegmental area but also from the cerebral cortex and thalamus and performs an integration of rewards and actions [36]. If the individual's sense of self-determination [37] of the method is strong, work performance is high and the activity of the ventromedial prefrontal cortex is also high, regardless of the success or failure of the task [38]. Anxiety and fear consume processing resources in the central executive system (dorsolateral prefrontal cortex), therefore, stress reappraisal provides a margin to behaviour [39]. The cerebellum, which controls unconscious behaviour, is connected to the reward system [40]. There is a dopamine transmission mechanism that emphasises rewards rather than the prediction of effort [41,42]. These facts form the neurophysiological basis of what people like to play, even if they are inefficient and troublesome to others. In summary, behaviour is meaningful to individuals. It is difficult to define play universally based on behaviour. Similar to behaviour, play should not be defined by tools or methods. In addition, to promote the activation of the reward system, decision-making and proactive action (action based on will) are important.

### 2.3.2. Definition of Play

Addiction and gambling cannot be ignored when considering play. Addictive drugs act directly on the ventral tegmental area and nucleus accumbens and are associated with rewards. Many addictive drugs significantly increase the activity of the dopaminergic system, particularly at the terminal level of the nucleus accumbens [35]. It is well known that damage to the orbitofrontal cortex leads to a preference for high-risk, high-reward

gambling [43]. Therefore, it is not appropriate to consider a state in which brain function is abnormal due to addiction or trauma as play. Additionally, bonobo play is exclusively an action on other individuals and may include sexual activity [44]. A dopamine-high state is observed in individuals with various addictive behaviours such as gambling, sexual intercourse, and exercise [35]. The activation of the dopamine system is the root of play; however, uncontrollable excessive willpower should be ethically underestimated, as in human play. In addition, the life activities and maintenance of the species, such as life and death, reproduction, eating, sleeping, breathing, and digestion, are not important. These are not appropriate for inclusion in the definition of play, even if they involve the autonomic nervous system or dopamine system.

In this study, play was conceived as an internal state of the body, including the brain. It was defined as follows: play is the activation of one's reward system through intrinsically motivated decisions and actions of the self and not for the direct purpose of survival. In the narrow sense, it specifically refers to this act.

### 2.4. The Suitability of the Current Definition

We applied the above definition of play to certain life activities and examined whether it is possible to explain them from the viewpoint of intrinsic motivation, decision-making, action, and reward system activation.

First, we examined what has been considered play since ancient times, namely, wordplay, martial arts and other sports, musical performance, and games, in reference to Huizinga [2]. Second, we explored activities that are not usually considered play, such as consumption of food, sexual intercourse, addictive drugs, and work; and finally, we investigated gambling, the consumption of alcohol, artistic or creative activities, watching movies and videos, and bringing others together, as obscure activities that have in some instances been considered to be play. Although it seems that there cannot be a strict definition of "play," we have still offered a basic definition to examine its suitability.

### 2.4.1. Activities That Were Considered to Be Play

- Wordplay

There are two sides to oral communication: when words are uttered and when words are heard. When our speech is characterised by quick, witty comments, this may activate our reward system. Actively uttering or describing words are more playful experiences than simply thinking of words in the brain. The game constitutes play if the reward system is activated by seeking the words of others through intrinsic motivation, listening to them, or watching their writing or reading. Moreover, when two-way communication is achieved, activating the other person's reward system leads to activation of one's own reward system through empathy. In both cases, if the motivation was extrinsic or not based on one's own decision-making (e.g., being told or heard by others), it was not considered play.

- Sports and musical performance

The reward system is activated by one's own actions, but in the case of extrinsic motivation (e.g., aiming to obtain a high ranking in a competition), it is not considered play. In situations such as competing against strong rival teams or high-level tournament players, where coping resources are expected to outweigh the stress involved in the task, it can be viewed as a challenge, and a reward can be expected upon successful completion. By contrast, such a task may not be considered play if it is perceived to be too threatening, has no reward, and if the outcome is unpredictable.

- Games

These activities are usually defined as play. However, if any one of these activities is prompted by true extrinsic motivation (e.g., for salary purposes), behaviour without self-determination (e.g., boring learning games), or if the reward system is not activated (e.g., the enemy hits the game and one cannot expect to win at all and it cannot be reappraised), it ceases to be play.

### 2.4.2. Activities That Are Not Usually Considered to Be Play

- Consumption of food

The quest for good nutrition is not considered play since the motivation is extrinsic. However, in the case of the consumption of food at barbecues and parties, the purpose is a mixture of survival through eating and non-survival through celebration feasting, and so, it constitutes play in settings where pleasure is derived through acts of eating with intrinsic motivation for purposes other than nutritional intake.

- Sexual intercourse

The reward system may be activated by engaging in sexual intercourse if participating in the act is based on a self-determined choice through intrinsic motivation. However, when engaged in for the purpose of breeding, it is not considered play. In addition, if the decision-making is left to others, or if the emphasis on safety is extrinsically motivating, it is not considered to be play. Moreover, it is believed that if empathy for another person and empathy from another person is high, the reward system can be activated; that is, a partner's sense of reward can be felt by their partner if they share enough empathy.

- Addictive drugs

The use of addictive drugs and the associated pleasure does not constitute play since the act directly intervenes in the dopaminergic system, making the resultant enjoyment abnormally derived. In addition, as normal decision-making is hindered, even if the reward system is activated by one's own actions, it is not considered play. However, if activation of the reward system is expected by intrinsic motivation before ingestion, it could be considered to be play if certain conditions are met.

- Work

General work activities do not serve the purpose of survival. If such activities are performed with extrinsic motivation and follow the will of others, the reward system is not activated, unless it succeeds. In other words, this type of work does not constitute play. However, if the task is engaged in by self-determined choice and a sense of accomplishment results, the reward system may be activated through one's own actions, regardless of success or failure. If this type of work is based on intrinsic motivation, then it may be considered as play.

### 2.4.3. Obscure Activities That Were Considered Play in Certain Instances

- Gambling

Financial gain is generally an extrinsic motive, but when the collection of money becomes an intrinsic motive, it may constitute play. However, if normal decision-making is disrupted by the excessive activation of the dopamine network, engaging in this activity will not be considered to be play.

- Consumption of alcohol

If the reward system is healthy and not addictive, and if the reward system activation is based on intrinsic motivation rather than consumption for survival, then it may be considered as play. If the consumption is forced by others and does not depend on one's own decision making, it will not be play.

- Artistic or creative activities

These activities are often considered play since they are based on intrinsic motivation, decisions to engage are usually self-determined, and the reward is the satisfaction that is experienced as a result of one's actions. However, when we raise our expectations and set unrealistic goals for ourselves, and our stress levels exceed our coping resources, or when we are not satisfied with the product of our efforts or there is no reward, then the activity ceases to be considered play. When requested to engage in artistic activities by others for financial gain, the motivation becomes extrinsic, and it can no longer constitute

play. In addition, when the creative activity is limited and the sense of self-determination is curtailed (e.g., when one feels they are limited by their own capabilities), it is no longer considered to be play.

- Watching movies and videos

If an individual's reward system is activated by watching their favourite movie after engaging in this activity based on intrinsic motivation and they project the decisions and actions of the characters onto themselves through empathy, then this may be considered to be play. However, since this is not a direct physical experience of the self, it is not extremely intense.

- Bringing others together

For example, if a sports team manager assembles a group of players based on intrinsic motivation and guides the players through their own decision-making processes, which activates their own reward system, this may be considered as play.

Some life activities are explained by physiological definitions, based on brain mechanisms, rather than conscious and behavioural expressions. It was possible to explain what was previously considered to be play by delving deeper into the boundary between play and non-play and to provide an explanation for activities that had not yet been investigated within the field of play research. In addition, ambiguous activities can be explained without difficulty. These explanations are examples, and the characteristics of play in all activities can be generalised and described by viewing it in a manner similar to that of intrinsic motivation, decision making, action, and reward system activation.

*2.5. Scale to Assess Play*

Play is probably neither entirely evident nor nothing at all both among individuals and groups. Although concise expressions were used above to explain the suitability of the definition, play is considered to have a continuous intensity, from situations that do not constitute play at all to situations that are very playful. Therefore, as a method to continuously evaluate the degree of play, a scale based on the above definition was developed. The four elements are intrinsic motivation, decision-making, action, and reward system activation. Each element comprises a number of sub-items based on its functional meaning (Table 1). The sub-items were first measured using a Likert scale or Visual Analogue Scale (VAS) method. Next, the numerical values of the sub-items were averaged to obtain the mean score for each element, and the sum of the values was used to determine the degree of play.

**Table 1.** Playability scale.

| Element | Sub-Item [1] | Functional Meaning |
|---|---|---|
| Intrinsic motivation | 1. I always want to do it | Short-term motivation |
| | 2. I want to keep doing it forever | Long-term motivation |
| | 3. I want to do it everywhere | Independent of extrinsic factors |
| | 4. I want to do it more than anything else | Comparison with extrinsic factors |
| Decision-making | 5. If I am going to do it, I decide to do it myself and get to work | Sense of self-determination |
| | 6. I am willing to do it, even if others do not tell me | Decision-making initiative |
| Action of the self | 7. I use fully my body and tools to do it | Acting on Intent |
| | 8. I feel like I am doing it | Agency of action |

**Table 1.** *Cont.*

| Element | Sub-Item [1] | Functional Meaning |
|---|---|---|
| Reward system activation | 9. My heart feels good when I do it<br>10. My body feels good when I do it<br>11. Doing it makes me feel high<br>12. When I am doing it, I cannot care about anything else | Spiritual reward<br>Physical reward<br>Dopamine System<br><br>Addictiveness |

[1] The average of the sub-items is the element point, and the sum is the playability.

### 2.6. Trial of Evaluation Based on Playability Scale

In March 2020, an online survey was conducted to assess the degree of play and gain insight into what our sample population considered to be their favourite play activity. We also gathered information regarding activities participants did not like but continued to take part in anyway ("highest-effort" habits). The data were obtained from 824 men and women in their 30s and 40s who were married and had children in the Tokyo metropolitan area of Japan, for a total of 1648 people. The mean age and standard deviation of respondents was 39.98+/−5.76 years. The research monitors, registered with Macromill, Inc. (Tokyo, Japan), a research company in Japan, were potential respondents on the Internet. This study used Macromill's non-connectable anonymised dataset. The survey provided participants with sufficient explanations about their guaranteed anonymity, why the study was conducted, how the data were used, and whether there were any risks and allowed them to abandon their responses midway. The answer was an expression of a consent form. The sub-items in Table 1 were answered on a Likert scale ranging from 0 (not applicable) to 4 (true). Therefore, the degree of play ranged from 0 to 16. To prevent bias, the terms well-being and enjoyment were never used.

### 3. Results

The results indicate that the participants' favourite play activities were games (16.3%), sports (11.0%), gambling (5.8%), watching movies and videos (4.8%), karaoke (4.1%), travelling (4.1%), playing smartphone games (3.9%), and reading (3.9%). These were followed by driving, cycling, shopping, and the consumption of alcohol. The highest-effort habit or activities that they did not like to do but continued to do anyway were work (23.1%), housework (9.4%), exercise (7.3%), walking or jogging (6.9%), and cooking (6.7%). Following these activities, the next highest-effort habits were cleaning and studying. These raw data are available at Supplementary Materials S1. Figure 1 shows the percentage of responses for all sub-items. The results of the paired *t*-tests for these element points and the total points are shown in Figure 2. The favourite play activity showed a significantly higher degree of play than the highest-effort habit. There was a significant difference in intrinsic motivation, reward system activation, and decision-making, but no difference in action of the self. In addition, Figure 1 shows that the sub-items fluctuated even within the element.

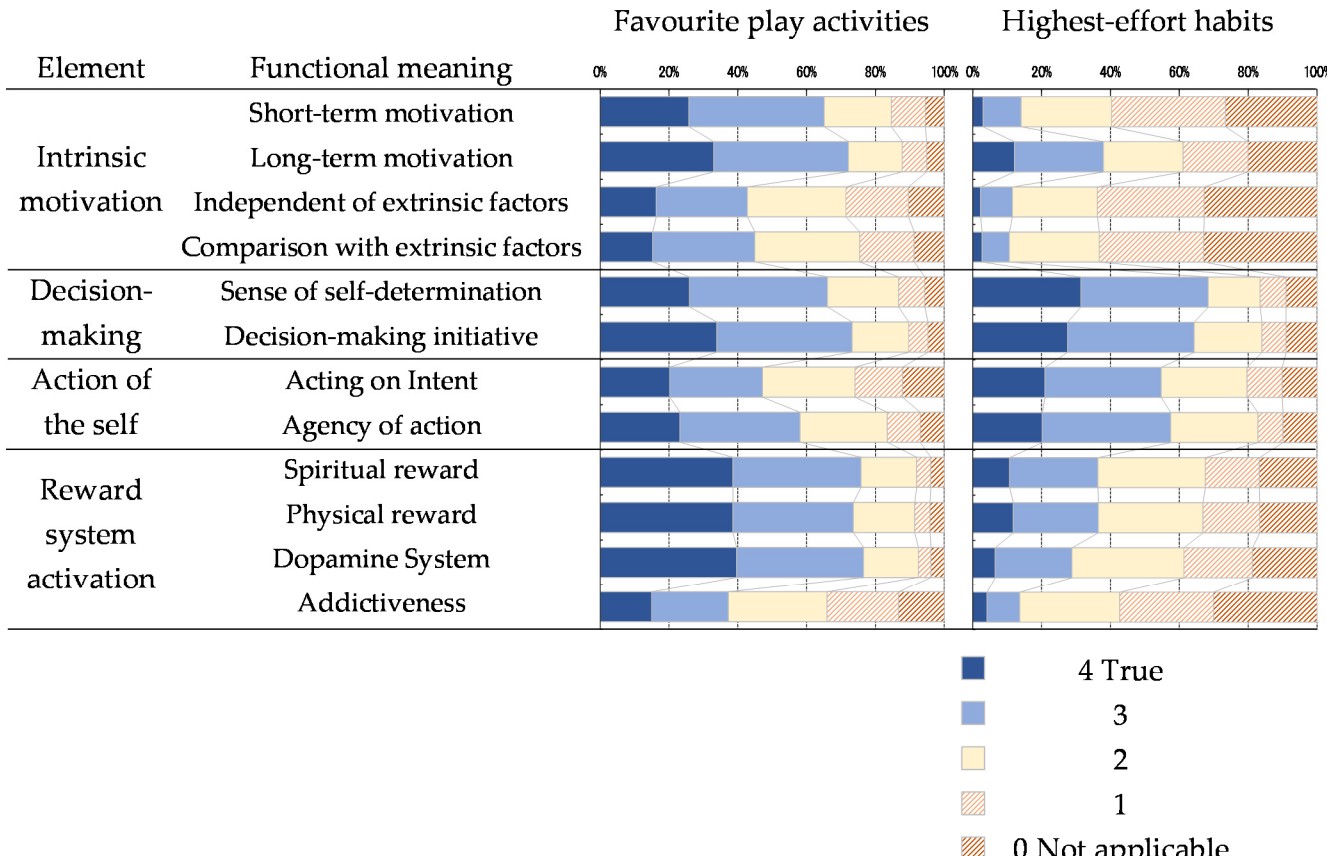

**Figure 1.** The favourite play activities (**left**) and the highest-effort habit (**right**) chosen by participants.

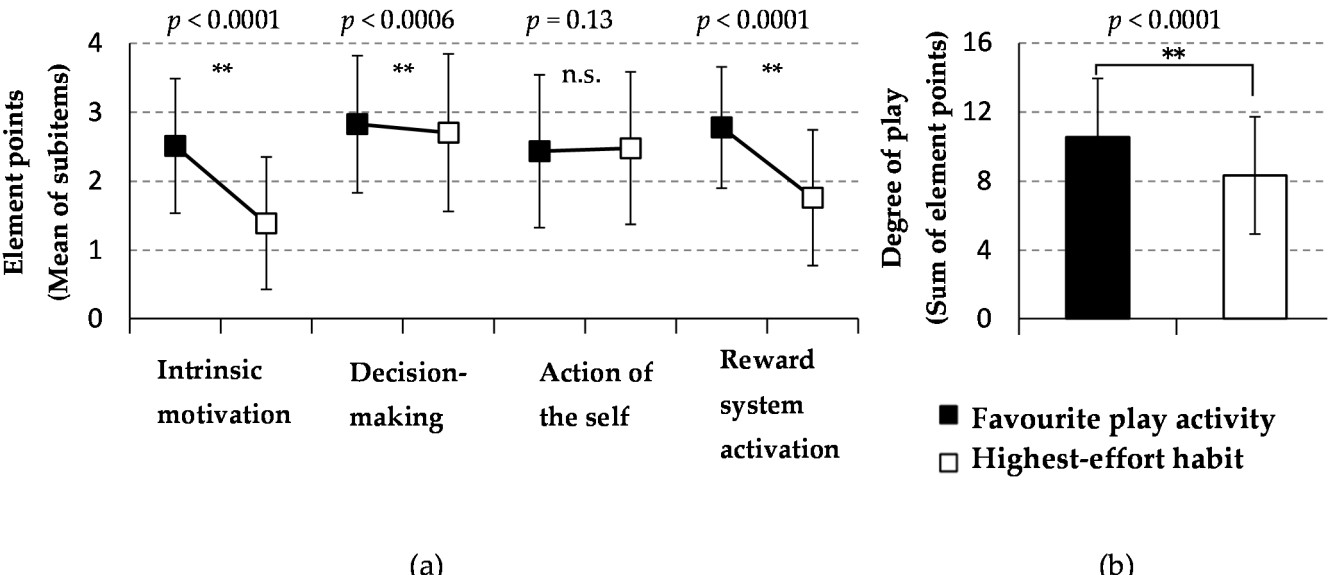

**Figure 2.** Results of the evaluation trial on the favourite play activities and the highest-effort habit: (**a**) Element points (ranging from 0 to 4); (**b**) Playability (range: 0 to 16). The filled and open symbol represent the favourite play activities and the highest-effort habits, respectively. Mean value ± SD. $p < 0.01$ with **.

## 4. Discussion

The definition of play in this study can be explained consistently regardless of whether it is formally adopted in the conventional field of play research. This consistency is important in the natural sciences [45]. The attention was not focused on the degree of expression of consciousness and behaviour but rather on intrinsic motivation, decision-making, action of the self, and reward system activation that appear to be at the root of these facets. As a result, it was possible to explain the state of non-play in what was conventionally considered to be play (e.g., games) and the state of play in what was conventionally considered not to be play (e.g., work). Therefore, our definition of play is more universal and comprehensive than the conventional definition. Furthermore, applying a scale based on this definition made it possible to refer to the brain mechanisms that are associated with play activities. In the evaluation trial on a playability scale, the temporal variable was the largest in the element of intrinsic motivation for favourite play activities. However, extrinsic factors are shown to reduce intrinsic motivation. In other words, if there is a way to maintain intrinsic motivation, this type of motivation may be enhanced. In addition, the highest-effort habits were generally low in the sub-items of intrinsic motivation and only high in long-term motivation. This reflects the long-term expectation that activities such as work, housework, and exercise, which have risen to the top of the list of the highest-effort habits, lead to an increase in the value of one's family and self-worth. While the decision-making factors were significant, the absolute difference between favourite play activities and the highest-effort habits was significant but small. This shows that decision-making in an efforted habit's top job and household was as strong as that of gaming, sports, and gambling, which were found to be the favourite play activities. There were no differences in the elements of action between the conditions. This may be partly due to a relative decline in physical and tool mastery under high goal setting in favourite play activities and sufficiently high work and housework skills in the highest-effort habits. Our findings also support the difficulty of explaining play based on behaviour alone. The reward system activation factor was remarkably higher in all sub-items for favourite play activities. The low degree of addiction suggests that there was no excessive activation of the reward system beyond the voluntary control in the sample population. As mentioned in Section 2.4.3, if the activity is driven by intrinsic motivation and activates the reward system by one's own actions, gambling can be considered as play. However, if decision-making is not normal, and the inhibitory system in the brain does not function adequately, it is no longer classified as play. Although fMRI (functional Magnetic Resonance Imaging) evidence is needed to determine whether the condition has occurred, it is important to note that this study does not intend to provide evidence. It lays the groundwork for future research. The hypothesis is that individuals with a high degree of playability when initially engaging in activities like gambling or drug abuse may not be able to quit and this may later lead to addiction. The indicators of this study are believed to facilitate an examination of this phenomenon.

The current method is applied as follows: to promote the well-being of a group in a challenging and harsh work environment, workers rate their playability. If the cause of the decrease in playability is a decision-making factor, providing options may lead to improvements in playability. To promote the purpose of life of individuals in an organisation, they first evaluate their degree of play in various activities. The results showed that the goals of work assignments and welfare will be settled based on activities that have higher intrinsic motivation and reward system activation. Another point of view is that if all aspects of playability between partners are high, their sexual lives are good. If one of the partners possesses a low-level degree of play in sexual intercourse–for example, a low element point for the decision-making of only one of them–something is wrong. In other words, well-being related to sexual issues could be evaluated. Thus, in general, by quantitatively comparing the elements of play in each domain in this way, it can be used as a reference for the development of policies to enhance well-being. The domains to be

compared include type of activity, ethnic, regional, individual, gender, generational, and experience differences.

Proyer was concerned that the silent variants of playfulness (e.g., childlike whimsy) would be subject to aversion as well as laughter or bullying others [18]. Thus, the study of play does not always focus on its positive aspects. In this study, the terms degrees of play and playability were used. The use of words by someone who is less playful or who is suffering can be an object of disgust. In such cases, the use of other terms such as physiological playfulness is recommended.

## 5. Conclusions

In this study, we defined play using what is known of the brain mechanisms common to humans. Play is the activation of one's reward system through intrinsically motivated decisions and actions of the self, not for the direct purpose of survival. In the narrow sense, it specifically refers to the act of play. The OECD's 2011 report states that although various well-being indicators can be used to paint a broad picture of people's lives, the measurement of well-being remains challenging [1]. In the future, the evaluation method proposed in this study will make it possible to examine well-being more broadly. For example, by identifying factors that reduce the degree of play at work and using them to improve well-being and identifying jobs that do not balance playability (rewards) and wages. The method may also be used as an auxiliary method for detecting signs of addiction to activities such as gambling. It also provides practical advice for evidence-based countermeasures, such as suggesting play activities as an alternative to harmful addictive activities or substances. This is the basis for incorporating elements of play into education to motivate students in schools. The design of a working environment is an evaluation criterion other than the work performance, which is often the goal of ergonomics. It is one of the materials that examines whether well-being is protected during the pandemic and harsh life. By incorporating playability into the items of the well-being survey, the accuracy of the survey is enhanced, contributing to the accountability of differences between nations. Our study aimed at exploring well-being holistically by defining play as an internal state involving the body and the brain. It serves as a historical foundation and informs future development regarding the utilisation of play as a sustainable way to fostering well-being and, in some cases, health. This transcends age, race, geographic location, or other demographic distinctions. To validate this notion, it is necessary to implement this methodology in diverse cultural spheres and domains. The study did have some limitations. The scale used in this study was not directly compared with that of other studies. Neither was the study tested for multicollinearity between questions nor verified for the validity of the scale itself. In addition, it has not been examined in addicts or people with extreme life difficulties. When using the playability scale for people with diminished well-being, the term playability may be criticised and this may require further assessment before implementation.

**Supplementary Materials:** The following supporting information can be downloaded at: https://www.mdpi.com/article/10.3390/su151310725/s1, Supplementary Materials S1 representing the favourite play activities and the highest-effort habits from our survey.

**Funding:** This research was funded by TOKYO GAS Co., Ltd. grant number J19KK00185 under a joint research agreement. The APC was funded by Chiba University.

**Institutional Review Board Statement:** Not applicable. The research monitors, registered with Macromill, Inc., a research company in Japan, were potential respondents on the Internet. This study used Macromill's non-connectable anonymised dataset.

**Informed Consent Statement:** Informed consent was obtained from all subjects involved in the study.

**Data Availability Statement:** The data presented in this study are available on reasonable request from the corresponding author. The data are not publicly available due to privacy restrictions.

**Acknowledgments:** The author would like to express gratitude to Shinichi Kagiya and Aki Kawamura for their assistance in administering the survey and for their advice.

**Conflicts of Interest:** The author declares no conflict of interest. The funders had no role in the design of the study, collection, analyses, interpretation of data or the writing of the manuscript, or in the decision to publish the results.

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
