# Peer review of "The Definition of Play: A Measurement Scale for Well-Being Based on Human Physiological Mechanisms"

_sustainability, doi:10.3390/su151310725_

Round 1
Reviewer 1 Report
Thank you for the possibility to review the article entitled „The definition of play: A measurement scale for well-being based on human physiological mechanisms”.
Although the author's idea to create a new tool for the study of psychological variables was good and worthy of attention, unfortunately, the execution and preparation of the article have definitely many limitations for it to be published in a scientific journal in this form. Both the title and the content of the article prove that the author uses words such as play, well-being, enjoyment, and pleasure interchangeably (perhaps even as synonyms). From a psychological point of view, I find it difficult to agree with the author's approach. In psychology, there are clearly defined definitions of what is well-being and what is fun, and I was absolutely not convinced by this scale, which the author called the playability scale.
What is the author really examining with his own tool - playability, source of these expressions of enjoyment, well-being?? I have the impression that the author is not entirely convinced what the construct he created is, what is it a measure of?
Allow me to expand my opinion:
Line 35-37 - “In general, the role of enjoyment in well-being has been taken for granted and has not been emphasised”. What does the author mean?? this is not a clear statement for me
Line 38-41 - The author suggests that it is necessary to review the research on well-being, he draws attention to "various domains" such as loneliness, transportation, young people - Are you sure these are the domains of well-being??? What aspects of well-being were measured in these studies?
Line 74-75 - “For example, playfulness in adolescence has been shown to be linked with various domains, such as academic performance, anxiety, and bullying, which are specialized domains of well-being [20]”- Are you sure anxiety and bullying specialized domains of well-being?? Now (according to the author) the content suggests that bullying is conducive to well-being and is its predictor (we know that it can be a negative predictor, but it was not mentioned in the text).
The review of the literature on the understanding and definition of play is presented by the author in the Materials and Methods section. Why?
Line 123-134 - The author relies on scales examining various constructs. These scales do not measure play but well-being, quality of life, life purpose, and happiness - these are separate psychological constructs related to play and pleasure, but they definitely do not measure the same construct (and none of them has the aspect of play, but they do have pleasure).
Line 132-133 - “However, these studies measured the results of play, such as psychology and behaviour, through subjective observations and did not measure play itself”. - I don't understand this sentence at all. What does it mean according to the author that the result of the play is behaviour or psychology? What psychology? How did that not measure play? The tool proposed by the author also does not measure play! The method of examining the source of manifestations of joy is not the same as the well-being scale or the scale of play itself.
Similarly, in line 166 - "Behavior and psychology are the two most significant aspects of play research" - What does the author mean when he writes that psychology is an aspect of play?? What is this aspect of psychology in play? After all, it is in psychology that we distinguish, for example, different stages or types of play based on cognitive or social development.
Line 137- 138 - “It is necessary to elucidate what play means to human beings and upon investigation we have discovered that the source of the expression of behaviour related to play is found in the brain”. Who is the author of such a discovery?
The great advantage of this text is the author's reference to neuropsychology and an attempt to combine specific functional mechanisms with the characteristics of experiencing pleasure from performing certain activities (but in my opinion, these are not elements of play).
The author created a definition that also applies to neuronal activation. “In this study, play was defined as follows: play is the activation of one's reward system through intrinsically motivated decisions and actions of the self, and not for the direct purpose of survival. In the narrow sense, it specifically refers to this act”. How then did the author investigate whether the reward system was activated through intrinsically motivated decisions? Did the author ask the respondents whether each of the pleasures mentioned by them was internally motivated? Did the author assume the hypothesis that since this action gives pleasure - it is always caused by intrinsic motivation?
The author in the text discusses which activities can and should not be included in the play, but then in his own research it is difficult for the reader to determine on what basis the author decided that, for example, gambling is play, not addiction. How did the author check whether it was just "play" or "compulsion, addiction”?
It is difficult to agree with the author's assumption that travel should be treated (and it is by the author of the article) as a play activity. Why is travelling considered play? How did the author distinguish play from a hobby or a habit?
The criteria of what according to the author is play included in the results are unclear to me.
In my opinion, the closest construct to what the author actually studied is favourite activities or activities that are a source of manifestations of joy, giving pleasure but I find it hard to consider that is play.
I have doubts about how the author separated what is a source of joy, e.g. in games, from addiction in the subjects. Was it controlled in any way in the studies?
My doubts are raised by questions from the first subscale - which the author called intrinsic motivation and shows the desire to have fun, but on the other hand (psychological) such questions may well be indicative of addiction - ("strong desire to do something wherever I am" - and do not necessarily have to concern they have fun).
The fact that I enjoy, for example, playing or watching TV does not necessarily mean that "I want to do it everywhere".
My next objection concerns the very construction of the questionnaire. Did the author check whether the scale is reliable and valid? Does the scale really test what the author assumed, or do these made-up questions combine into the subscales proposed by the author? What are the psychometric properties of this tool?
On what theoretical basis did the author base the assumption that the statement “my heart feels good when I do it” is a spiritual reward? Why, in the author's opinion, the statement - “Doing it makes me feel high” will be related to the dopamine system.
From my point of view, item 12 is also ambiguous, because the statement "When I am doing it, I don't care about anything else" is assigned to functional meaning - addictiveness and why not mindfulness?
In the description of the results, the author uses yet another name for what was the subject of the research - favourite play activities.
The author does not convince me with a too broad definition of play (treated as a favourite play activity), claiming that each favourite activity like sport or travel should be treated (according to the author) as a favourite play activity. If not, where is the line between an ordinary activity that gives us pleasure and we like to do it and fun? Because I did not see this border in the reviewed text, unfortunately.
Another problem is the lack of a described group of respondents, and the lack of a described research procedure. Who were the test persons? What were they asked about? What additional variables, e.g. demographic, were taken into account (age, education, place of residence, gender, etc.)
How were the respondents surveyed? Where? how long did the study take? How many subjects wrote down pleasant and unpleasant activities? How did the author calculate these activities? where did the percentage results come from? Why are there no raw results?
Despite the interesting topic presented by the author, in my opinion, the article requires an additional revision by the author and a clear determination of what was the subject of the study and what the constructed tool concerns. In my opinion, this tool is strongly theoretical, although poorly embedded in theory, especially psychological, with unknown psychometric properties.
I encourage the author to make a major revision of the text and resubmit it to the journal.
Of course, the author has the right to disagree with my point of view and be convinced of the rightness of his idea. So please treat my allegations as suggestions resulting from the point of view of a researcher whose knowledge is firmly embedded in the discipline of psychology.
Reviewer 2 Report
Changes
The paper is very interesting read.
There are certain suggestions which may help author to improve the following manuscript.
1. The author is linking wellbeing with play through comparing various activities. The concept of wellbeing has been defined in varied way in the research literature. The current concept of play or activities can affect both subjective wellbeing as well as psychological wellbeing. It will be really good if author can think of this and can specify or link play with types of wellbeing. Even author can define or adopt a particular wellbeing to avoid confusion.
2. Line- 18 which method author is talking about. Here the abstract can be little more structured as how the author has established the definition and how the online survey has provided evidence to it. the explanation of little method or change in overall structure of abstract can be done to provide reader a clear understanding about the procedure taken up by the author to establish the definition.
3. Line 14 to 15 - a differentiation can be made throughout the paper about either play versus non play activities or play as spontaneous versus highest effort habits. use any two or three similar terminologies throughout the paper to avoid confusion
4. in line 28 author is trying to differentiate well-being with enjoyment. here explanation of wellbeing type can give a better clarity
5. Line 54 when we are calming common to all human societies, are we showing enough data to support it. as the activities showed as play in current work does not represent many cultures. similarly if activities are different there could be some different aspect which may miss. author here can be grounded more in context of data collected.
6. Line 131 is unclear. Author wants to say about method of measurement or method of assessing the impact of play.
7. Line 132 and 133 is unclear. As author is stating that psychology and behavioural outputs cant be holistic to understand play. However in line 138 author is supporting same.
As per my understanding here author may say that mechanism to explain the effect to this date haven’t been explored therefore author using so and so theory or concept presenting a more unified expression and connection between play and wellbeing.
8. As the scope of the journal is sustainability. The current article can connect the concept of sustainability with play in various ways. The conclusion part can be used to set up a connection as how the historically the concept of play regardless of age, race, geographical locations or other demographic differences, concept of play has been utilized as sustainable method for promotion of wellbeing and in some instances health too.
9. Line 166: the word psychology denotes a subject however behaviour is one component of this subject. The author rather than using word psychology can find more appropriate and specific constructs of psychology which is used in play literature
10. Line 398 and 399 the word psychology is creating confusion
11. Throughout the article author is stating that a different method here is used in play research. However, it is not clear what method means here. It seems that more of the different theoretical stance may be used here then method. As quantitative and Likert scale is used similar to other previous research.
12. The author from line 426 to 439 tries to show the application of current scale in understanding and implementing the results in real word. However, my suggestion is here, the author in intro set up a challenge in play research and suggest that how the current method will be used in certain setting. Using the same setting challenged in intro at the end will help reader to understand the implications much better.
13. The heading “definition of play” can be reframed or divided in to two “proposed model”/ “biological basis or model” and then final definition
14. Overall the article is very unique. The author here tried to give a deeper and more detailed framework to link well being with play regardless of setting, which is very interesting part. Inclusion of images or some proposed model figures can give reader a better understanding about the proposed model.
15. A little more structured section on the quantitative study can also be included. As the demographics of the participants here can help author to ground the data in to specific context.
Round 2
Reviewer 1 Report
Dear Author,
Thank you very much for your answer. The author's arguments in many responses to reviews convinced me and showed an anthropologist's perspective.
The only suggestion I would like to make to the Author to improve the manuscript further is:
I think that the discussion of the results should include the answer that the author gave me in connection with the distinction between play and addiction:
“However, if decision-making is not normal, and the inhibitory system in the brain does not function adequately, it is no longer classified as play. Although fMRI evidence is needed to determine whether the condition has occurred, it is important to note that this study does not intend to provide evidence. It lays the groundwork for future research. The hypothesis is that individuals with a high degree of playability when initially engaging in activities like gambling or drug abuse may not be able to quit and this may later lead to addiction. The indicators of this study are believed to facilitate an examination of this phenomenon”.
In my opinion, it is essential that such a piece of clear and legible information be read by every reader because it shows the importance of the research carried out by the author.